# A morphotropic phase boundary in MA$_{1-x}$FA$_x$PbI$_3$: linking structure, dynamics, and electronic properties

Tobias Hainer, Erik Fransson, Sangita Dutta ⓘ, Julia Wiktor & Paul Erhart ⓘ ✉

Understanding the phase behavior of mixed-cation halide perovskites is critical for optimizing their structural stability and optoelectronic performance. Here, we map the phase diagram of MA$_{1-x}$FA$_x$PbI$_3$ using a machine-learned interatomic potential in molecular dynamics simulations. We identify a morphotropic phase boundary (MPB) at approximately 27% FA content, delineating the transition between out-of-phase and in-phase octahedral tilt patterns. Phonon mode projections reveal that this transition coincides with a mode crossover composition, where the free energy landscapes of the M and R phonon modes become nearly degenerate. This results in nanoscale layered structures with alternating tilt patterns, suggesting minimal interface energy between competing phases. Our results provide a systematic and consistent description of this important system, complementing earlier partial and sometimes conflicting experimental assessments. Furthermore, density functional theory calculations show that band edge fluctuations peak near the MPB, indicating an enhancement of electron-phonon coupling and dynamic disorder effects. These findings establish a direct link between phonon dynamics, phase behavior, and electronic structure, providing a further composition-driven pathway for tailoring the optoelectronic properties of perovskite materials. By demonstrating that phonon overdamping serves as a hallmark of the MPB, our study offers insights into the design principles for stable, high-performance perovskite solar cells.

In recent years, halide perovskites have garnered significant attention for their exceptional photovoltaic properties. These materials exhibit long carrier lifetimes[1–3] and high defect tolerance[4,5], making them strong candidates for next-generation solar technologies. Their conversion efficiencies have exceeded 25%[6], rivaling those of state-of-the-art solar cells. Moreover, their tunable bandgap and compositional flexibility allow precise control over optoelectronic properties[7,8], enabling further enhancements in device performance.

Among the most promising halide perovskites are MAPbI$_3$ (methylammonium lead iodide) and FAPbI$_3$ (formamidinium lead iodide). MAPbI$_3$ has demonstrated high efficiency and ease of fabrication[9,10], making it a benchmark material for perovskite solar

cells. However, it suffers from limited thermal stability, undergoing phase transitions at elevated temperatures that degrade device performance[11]. In contrast, FAPbI$_3$ exhibits greater resistance to thermal decomposition[12,13] and has a slightly lower bandgap, making it an attractive alternative. Nevertheless, its phase stability remains a challenge, as it readily transitions into non-perovskite phases at room temperature, limiting its photovoltaic potential[14,15].

Mixing MA and FA cations to create MA$_{1-x}$FA$_x$PbI$_3$ has emerged as a strategy to combine the desirable properties of both materials. By tuning the MA-to-FA ratio, the structural stability, phase behavior, and electronic properties of the material can be optimized, achieving a balance between efficiency and stability[16]. This approach holds

Department of Physics, Chalmers University of Technology, Gothenburg, Sweden. ✉e-mail: erhart@chalmers.se

promise for developing perovskite materials that are both high-performing and durable under real-world conditions.

However, the introduction of mixed cations also adds complexity, and the phase diagram of the $MA_{1-x}FA_xPbI_3$ system remains incompletely understood. Experimentally probing these mixed phases has proven challenging, and no general consensus has been reached on their stability and transitions[17–21]. Yet, understanding these phase relationships is crucial for advancing the design of stable, high-efficiency perovskite solar cells.

Compositions rich in $MAPbI_3$ exhibit three distinct structural phases: a high-temperature cubic phase ($a^0a^0a^0$), a tetragonal phase with out-of-phase octahedral tilting ($a^0a^0c^-$), and a low-temperature orthorhombic phase ($a^-a^-c^+$). In contrast, $FAPbI_3$-rich compositions display two well-defined phases: a high-temperature cubic phase ($a^0a^0a^0$) and a tetragonal phase with in-phase tilting ($a^0a^0c^+$). Additionally, a low-temperature phase is observed, though its detailed structure remains uncertain[22]. Here, the Glazer notation[23], given in brackets, describes the tilt patterns of the $PbI_6$ octahedra.

A key observation is that the tetragonal phases of $MAPbI_3$ and $FAPbI_3$, which occur at room temperature, exhibit opposing tilt patterns ($a^0a^0c^-$ vs $a^0a^0c^+$) and belong to two different space groups (I4/mcm vs P4/mbm). Given the full miscibility of $FAPbI_3$ and $MAPbI_3$, this suggests the presence of at least one morphotropic phase boundary (MPB). In general, a MPB is a region in the composition-temperature phase diagram of a solid solution—typically in but not limited to ferroelectric or piezoelectric materials—where two crystallographic phases with different symmetries coexist or are nearly degenerate in energy. MPBs are well known in oxide perovskites and a key feature of many functional materials, particularly in ferroelectrics such as $PbZr_xTi_{1-x}O_3$, where they lead to enhanced piezoelectric and dielectric properties[24,25]. While extensively studied in oxides, MPBs have received little attention in the context of halide perovskites, where their influence on phase behavior and optoelectronic properties remains largely unexplored. Identifying and characterizing this potential MPB could provide deeper insights into the structural dynamics of mixed-cation perovskites, clarify their impact on device performance, and introduce a further route for tuning optoelectronic properties through compositional engineering.

In this study, we investigate the phase diagram of $MA_{1-x}FA_xPbI_3$ using a machine-learned interatomic potential and molecular dynamics (MD) simulations to systematically map phase transitions and explore the underlying atomic-scale dynamics. Our results reveal the presence of a MPB at ~27% FA, which separates the $a^0a^0c^-$ and $a^0a^0c^+$

phase regions. Interestingly, this boundary remains nearly invariant with temperature, indicating a robust compositional threshold between these structural phases.

We further analyze the role of soft (tilt) modes associated with these phases and find that they are heavily overdamped. Despite this, their signatures persist well into the high-temperature cubic phase ($a^0a^0a^0$), where remnants of these distortions remain detectable several hundred kelvins above the cubic-tetragonal phase transition. At low temperatures, our simulations indicate a single-phase region characterized by orthorhombic symmetry ($a^-a^-c^+$), further defining the stability of different structural regimes.

Finally, we demonstrate that the crossover in dynamics at the MPB significantly influences electron-phonon coupling, leading to a pronounced maximum in the fluctuation of the valence band maximum (VBM) level. This suggests that the particular soft phonon dynamics near the MPB may amplify dynamic disorder effects that are relevant to charge transport and optoelectronic performance. Our findings provide key insights into the complex phase behavior of mixed-cation perovskites and establish a framework for tuning their structural and electronic properties through composition engineering.

## Results

### A machine-learned interatomic potential for $MA_{1-x}FA_xPbI_3$

A computational study of the phase diagram of $MA_{1-x}FA_xPbI_3$ requires a model that is both accurate and efficient. Such a model must capture subtle energy differences between phases, account for the rotational dynamics of organic cations, and enable sampling of system dynamics across relevant length and time scales. To meet these requirements, we developed a machine-learned interatomic potential using the neuroevolution potential (NEP) approach (see the Methods section for details as well as Figs. S1, S2, and S3 of the SI).

Our NEP model achieves high accuracy, with root mean square errors (RMSEs) of 3.9 meV atom$^{-1}$, 82 meV Å$^{-1}$, and 134 MPa for energies, forces (Fig. 1a), and stresses, respectively (also see Fig. S4 of the SI). The corresponding correlation coefficients ($R^2$) of 0.9997, 0.9896, and 0.9938 further demonstrate its precision. Additionally, the model accurately reproduces the rotational energy landscape of organic cations (Fig. 1b–d) and shows low RMSEs for energies, forces, and stresses across various structural prototypes (Figs. 1e and S4), demonstrating its suitability for exploring the phase behavior of this system. Crucially, it is also computationally efficient, achieving a speed of up to $14 \times 10^6$ atom step s$^{-1}$ on Nvidia GPUs (including, e.g., the A100 and RTX3080 chips). With a time step of 0.5 fs, this corresponds to a

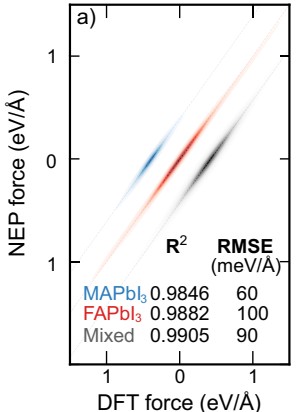
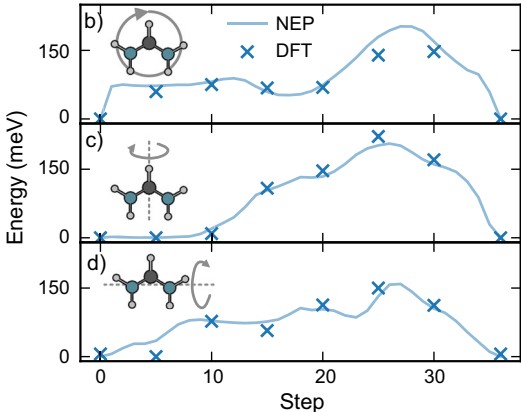
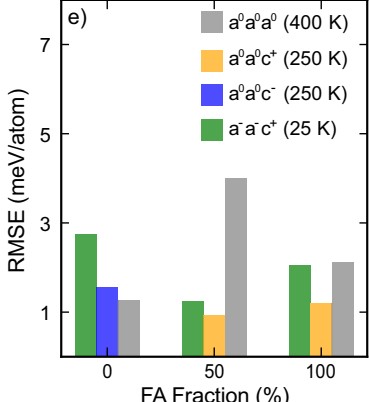

**Fig. 1 | Machine-learned interatomic potential for $MA_{1-x}FA_xPbI_3$. a** Force parity plot, categorized by structure type. The different structure sets have been offset along the x-axis for clarity. **b–d** Nudged elastic band (NEB) calculations along three rotational paths, each corresponding to the rotation of a FA molecule around a different axis, indicated by gray arrows. The crosses represent the energies calculated with density functional theory (DFT), while the continuous lines show the energy predictions of the neuroevolution potential (NEP) model along the different paths. **e** Root mean square errors (RMSEs) between NEP and DFT calculations for snapshots sampled from MD simulations across various phases and compositions.

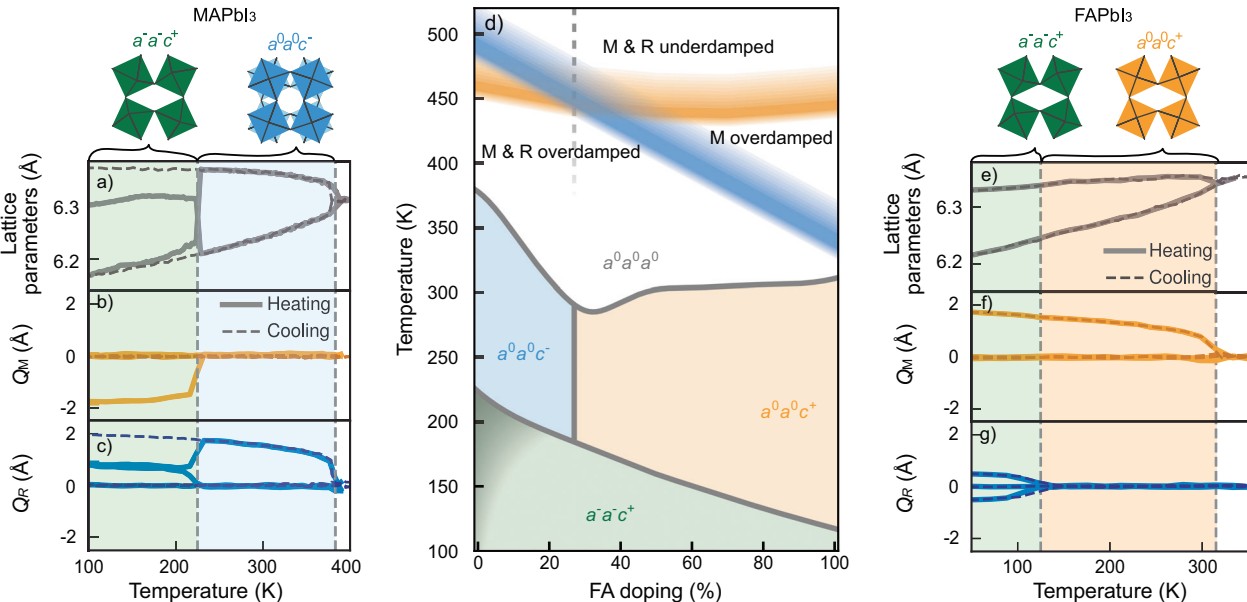

**Fig. 2 | Phase diagram of MA$_{1-x}$FA$_x$PbI$_3$ from atomic-scale simulations.**
**a**–**c** Lattice parameters and activation of M and R phonon modes during heating (solid lines) and cooling (dashed lines) of MAPbI$_3$. Dashed vertical lines indicate phase transitions. The tetragonal and orthorhombic phases observed during heating are illustrated by the PbI$_6$ octahedral tilt. **d** Predicted phase diagram based on phonon mode activation, labeled using Glazer notation. On the MAPbI$_3$ side the lower transition exhibits hysteresis due to its weak first-order character. The transition is therefore by a shaded area[27]. Regions of the cubic ($a^0a^0a^0$) phase are categorized as either overdamped or underdamped based on autocorrelation functions of the M and R modes. Note that the vertical line between the two tetragonal phases is the center of the MPB, and the dashed vertical lines is the mode crossover. **e**–**g** Lattice parameters and activation of M and R phonon modes during heating (solid lines) and cooling (dashed lines) of FAPbI$_3$, with tetragonal and orthorhombic phases illustrated as in MAPbI$_3$.

simulation throughput of 6.3 ns d$^{-1}$ for structures of 96,000 atoms, enabling large-scale molecular dynamics simulations.

## Phase diagram from mode projection

Initially, we explored the phases of the boundary compositions by performing heating and cooling MD simulations over a temperature range of 1 K to 400 K. From the resulting atomic trajectories, the displacements of iodine atoms were extracted and projected onto the R and M phonon modes of the cubic structure. This provided mode amplitudes, which were used to identify the tilt patterns and assign structural phases as a function of composition and temperature.

For MAPbI$_3$, phase transitions were identified based on abrupt changes in phonon mode activation, which correlated with variations in lattice parameters (Fig. 2a–c). At high temperatures, above ~385 K, the mode activation averages to zero consistent with a cubic $a^0a^0a^0$ structure. Between 385 K and 225 K, the R$_z$ mode was activated during both heating and cooling, indicating a tetragonal $a^0a^0c^-$ structure. Below this temperature, heating and cooling trajectories diverged. During heating, the system adopted the orthorhombic $a^-a^-c^+$ structure, the generally accepted low-temperature phase of MAPbI$_3$[26]. However, during cooling, the transition into the orthorhombic $a^-a^-c^+$ phase did not occur, consistent with a first-order phase transition with a large energy barrier[27]. This barrier is attributed to the required change in the tilt pattern of the third axis, from in-phase to out-of-phase tilting[27]. While sufficient thermal energy allowed the system to overcome this barrier during heating, on the time scale of the MD simulations, it remained trapped in a metastable state upon cooling. Thus, the observed transition temperature serves as an upper bound.

In contrast, FAPbI$_3$ shows nearly identical behavior during heating and cooling (Fig. 2e–g), as the sampled tetragonal-to-orthorhombic transition occurs between phases that are energetically more similar. The starting structure for the heating simulations was chosen as the orthorhombic $a^-a^-c^+$ structure[22], which is the same structure recovered on cooling. Three distinct structural phases were identified (Fig. 2e–g): the cubic ($a^0a^0a^0$) phase, where no specific phonon mode activation

was observed; the tetragonal ($a^0a^0c^+$) phase, characterized by M$_z$ mode activation; and the orthorhombic ($a^-a^-c^+$) phase, where R$_x$, R$_y$, and M$_z$ modes were all active.

By extending this procedure across the full composition range of MA$_{1-x}$FA$_x$PbI$_3$, a phase diagram was constructed (Fig. 2d). Across all compositions, cubic-to-tetragonal transitions occurred consistently during both heating and cooling, with structural changes following either the $a^0a^0a^0 \rightarrow a^0a^0c^-$ or $a^0a^0a^0 \rightarrow a^0a^0c^+$ pathways. Additionally, two tetragonal-to-orthorhombic transitions were identified: one between $a^0a^0c^-$ and $a^-a^-c^+$, and another between $a^0a^0c^+$ and $a^-a^-c^+$. The transition from $a^0a^0c^+$ to $a^-a^-c^+$ was observed in both heating and cooling simulations. However, the transition from $a^0a^0c^-$ to $a^-a^-c^+$ did not occur during cooling, indicating suppression, consistent with its first-order nature. Since this transition was only sampled during heating, the resulting transition temperature represents an upper bound. As a consequence, the precise temperature at which the orthorhombic phase becomes energetically favorable in this model remains uncertain. This uncertainty is indicated by the gray shaded region in the MAPbI$_3$-rich limit of the phase diagram (Fig. 2d). The lower bound of this region is guided by a previous free energy study of MAPbI$_3$, which identified the orthorhombic phase as the most stable structure below 90 K[27].

## Comparison with experiment

To validate our predicted phase diagram, we compare the observed phase transitions with experimental studies compiled by Simenas et al.[21] (Fig. 3). This comparison allows us to assess the extent to which our simulations capture experimentally observed trends and resolve discrepancies in reported phase behavior.

The high-temperature cubic-tetragonal phase boundary aligns well with experimental observations, deviating primarily by a nearly constant offset across the entire composition range[17,18,28]. For MAPbI$_3$, the cubic, tetragonal, and orthorhombic phases identified here are consistent with previously reported structures. For FAPbI$_3$, the phase diagram confirms the known cubic and tetragonal phases while

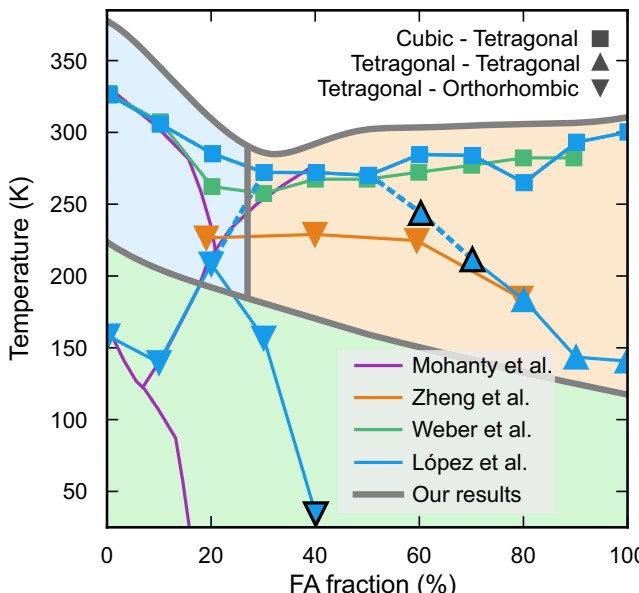

**Fig. 3 | Comparison of simulated phase boundaries to experimental results.** Data points are represented by markers, with their shapes indicating the crystal structures involved in the phase transitions. Markers outlined in black correspond to measurements with a limited data basis. The results from Weber et al.[17] predict a cubic-tetragonal boundary that aligns well with the findings of Francisco-López et al.[18], who in addition to this identify several transitions at lower temperatures. Zheng et al.[19] propose a tetragonal-orthorhombic boundary throughout the intermediate temperature range (note that these data are for nanostructures). Mohanty et al.[20] report phase boundaries similar to those in other studies but classify the intermediate phase as "large-cell cubic".

suggesting that the low-temperature phase adopts an orthorhombic $a^-a^-c^+$ symmetry reached on cooling.

Furthermore, our observations of the transition between two tetragonal symmetries, i.e., the presence of a MPB, are supported by experimental findings. Francisco-López et al.[18] observed a transition between two tetragonal symmetries at FA fractions 20% to 30% (see dashed blue line in Fig. 3). This agrees well with our placement of the MPB at the FA fraction 27%. Mohanty et al.[20] also identified a transition in this region, but instead of transitioning between two tetragonal structures they assigned a transition between a tetragonal and a "large-cell cubic" structure.

The intermediate composition range of MA$_{1-x}$FA$_x$PbI$_3$ below the cubic phase region has been challenging to probe experimentally. In this region, our simulations indicate the presence of tetragonal and orthorhombic structures. Based on temperature-dependent photoluminescence measurements on nanostructures, Zheng et al.[19] reported tetragonal-to-orthorhombic transitions at FA concentrations of 20% to 80%, consistent with the trends observed in this study. However, conflicting experimental findings exist for this region. The results from Francisco-López et al.[18] using photoluminescence and Raman scattering suggest that there are no orthorhombic structures above 40% FA fraction, instead assigning this region a possibly tetragonal symmetry and adding that there is disorder present. Note, however, that these assignments are not based on crystallographic analysis. Furthermore, an additional study using X-ray diffraction from Mohanty et al.[20] indicates a "large-cubic symmetry" for large parts of this intermediate region. These discrepancies highlight the challenges in experimentally resolving the phase behavior of mixed-cation perovskites, further emphasizing the role of atomistic simulations in providing complementary insights.

We note, however, that the results from Francisco-López et al. could be interpreted as supporting our findings. Their Raman

scattering results indicate clear phase transitions between tetragonal and orthorhombic phases for FA fractions 0% to 30% at temperatures between ~130 K to 210 K (indicated by the blue inverted triangles in Fig. 3). However, the measurement at FA fraction 40%, which suggests a similar transition near 40 K, is less conclusive (indicated with a black outline in the Fig. 3). A similar ambiguity exists for transitions on the FA-rich side of the phase diagram. Here, photoluminescence peak energy shifts have been interpreted as a transition from tetragonal symmetry to a phase described as disordered and possibly tetragonal, occurring between roughly 130 K and 200 K for FA fractions of 80% to 100% (blue triangles in Fig. 3). Transitions are also assigned for 60% and 70% FA. These are however not as clear, which is explained by the transition being continuous (marked with black outlines in Fig. 3). Excluding this less conclusive Raman and photoluminescence data, we find a tetragonal-orthorhombic phase boundary for FA fractions of 0% to 30% (130 K to 210 K) and a transition to a phase indexed as tetragonal for 80% to 100% (130 K to 200 K). However, the assignment of this tetragonal symmetry on the FA-rich side was tentative[17,29,30], and their neutron diffraction patterns did not allow for a complete classification of the unit cell symmetry. Given this, along with recent evidence suggesting that the low-temperature phase of FAPbI$_3$ is orthorhombic[22], it is reasonable to reconsider the classification of this FA-rich region as orthorhombic. This is also in line with the analysis of Zheng et al., which connects the two sides of the phase diagram through a tetragonal-orthorhombic phase boundary. So, if one takes into account the uncertainty associated with data points obtained by Francisco-López et al., it results in a phase diagram consistent with our simulations.

## Character of the morphotropic phase boundary

The phase diagram suggests a transition driven purely by compositional change—a MPB—between the $a^0a^0c^-$ and $a^0a^0c^+$ structures, corresponding to the vertical boundary between these phases (Fig. 2). Such a transition is inevitable in the phase diagram of this alloy, as the distinct phases of the boundary compositions cannot be combined without forming at least one MPB.

To investigate the MPB, we calculated and normalized the activation of the phonon modes along the primary tilt axis at 250 K (Fig. 4a). For compositions with FA doping significantly above or below the MPB, the system exhibited a strong preference for either the $M_z$ or $R_z$ modes, consistent with the expected $a^0a^0c^+$ and $a^0a^0c^-$ crystal structures. However, in the intermediate region, a smooth transition between the two modes was observed. The midpoint of this transition, corresponding to an FA doping of 27%, defines the center of the MPB.

The phonon mode projections provided detailed insights into the structural characteristics at the MPB. Analysis of the transition pathway between $M_z$ and $R_z$-dominated regions revealed the formation of alternating layers of either tilt pattern (Fig. 4a). These layers, aligned along the primary tilt axis, were nearly exclusively populated by either $M_z$ or $R_z$ modes, explaining the observed smooth transition. At the center of the MPB, an equal number of layers populated by $M_z$ and $R_z$ was present, marking the compositional crossover point.

As discussed in the next section, the emergence of these layers at the MPB is a consequence of the nearly degenerate free energy landscape for M and R modes when transitioning from the cubic to the tetragonal phase. Below the transition temperature, these modes freeze in, resulting in a layered distribution proportional to their likelihood at the transition.

Since the well-defined layered structures observed at the MPB could be influenced by the limited size of smaller simulations, we conducted large-scale simulations using systems of up to 2.6 million atoms. These systems were initialized in the cubic phase and cooled to temperatures where the MPB emerges. For MA$_{0.73}$FA$_{0.27}$PbI$_3$, two distinct regions were observed to form, stacked along the $\langle 011 \rangle$ direction, each exhibiting layered M and R mode activation (Fig. 4b). These

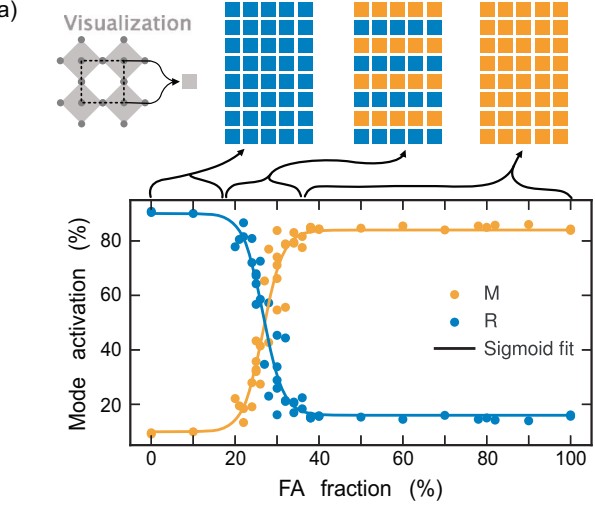

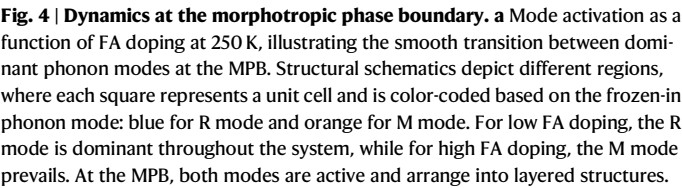

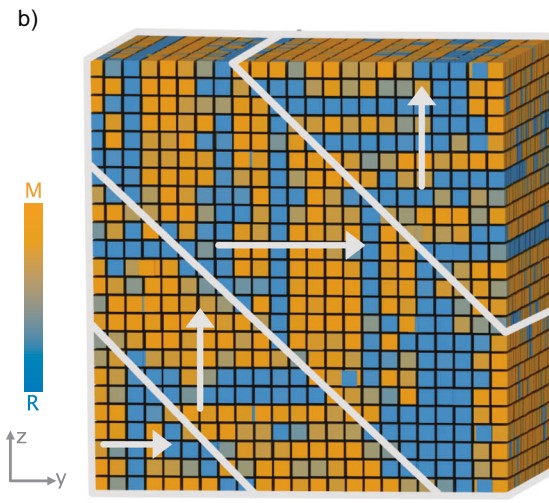

**Fig. 4 | Dynamics at the morphotropic phase boundary. a** Mode activation as a function of FA doping at 250 K, illustrating the smooth transition between dominant phonon modes at the MPB. Structural schematics depict different regions, where each square represents a unit cell and is color-coded based on the frozen-in phonon mode: blue for R mode and orange for M mode. For low FA doping, the R mode is dominant throughout the system, while for high FA doping, the M mode prevails. At the MPB, both modes are active and arrange into layered structures.

**b** Mode activation in $MA_{0.73}FA_{0.27}PbI_3$ from a large-scale simulation of 1.3 million atoms cooled to 260 K. Twinning behavior is observed, forming two distinct regions: one dominated by modes along the $z$-direction and the other along the $y$-direction, indicated by arrows. Within each region, alternating layers of M and R modes emerge, consistent with the layered structures identified in smaller simulations near the MPB.

regions were distinguished by a 90° rotation, with the modes within each domain aligned along alternating directions. The formation of different domains should not be interpreted as a finite-size effect, but rather as a consequence of the stochastic nature of nucleation pathways at finite cooling rates, and thus a finite-time effect. The coexistence of regions with different orientations primarily reflects the competition between local energy barriers for reorientation and the finite time available for domain wall motion, rather than a direct effect of system size itself. This is supported by the fact that for sufficiently large cells, one cannot observe a correlation between the size of the domains and the size of the simulation cell.

The layered structures and domain formations at the MPB point toward a structurally soft regime where the energy landscape between competing phases is shallow. Unlike classical ferroelectric perovskites, where ferroelectric, ferroelastic, and even ferromagnetic properties are maximized at the MPB[31], the present system does not exhibit intrinsic ferroelectricity. Instead, the presence of MPB-associated soft lattice dynamics may play a key role in optoelectronic properties as discussed further below, particularly through their influence on dynamic disorder and electron-phonon coupling.

The formation of 90° domain walls along the ⟨011⟩ direction, a feature previously observed in perovskites with ferroelastic and ferroelectric character[32,33], may suggest an intrinsic structural response to local stresses or instabilities. However, rather than long-range polarization, these domains likely reflect the ability of the system to accommodate structural frustration through dynamically fluctuating tilt patterns.

This complex domain morphology and the presence of internally modulated structural distortions further explain why experimental determination of the crystal structure of $MA_{1-x}FA_xPbI_3$ has been particularly challenging. Regions with different orientations, coupled with internal layers of frozen or fluctuating tilt modes, introduce a level of disorder that complicates conventional diffraction-based classification.

In general, when aiming to experimentally investigate MPBs, whether in $MA_{1-x}FA_xPbI_3$ or in other systems, it is important to recall the conditions necessary for their formation. Specifically, there must be distinct crystallographic phases at the boundary compositions as well as complete miscibility. With these criteria in mind, one can assess whether a given material is likely to exhibit a MPB. If so, this can serve as a basis for exploring possible domain structures and the coexistence of phases, as our work suggests that this could be good indicators for MPBs. Beyond structural considerations, we can also discuss strong electron-phonon coupling. The presence of soft phonon modes generally enhances the electron-phonon interaction, as the coupling strength scales as $1/\sqrt{\omega}$, where $\omega$ is the phonon frequency. This implies that the coupling should be stronger near a MPB, where multiple modes exhibit such soft behavior. As a result, the influence of the MPB on the electron-phonon interaction is expected to be significant. This is analyzed below in the section on electronic structure. Experimentally probing this behavior could thus provide further insights into the impact of MPBs on materials performance. One should, however, note that in contrast to systems with well-established MPBs such as $PbTi_{1-x}Zr_xO_3$, in the present case the two phases that meet at the MPB exhibit anti-ferrodistortive behavior and thus no spontaneous polarization. As a result, we do not expect a strong signature of the MPB in the dielectric response.

## Phonon mode dynamics

The dynamics of phonons play a fundamental role in determining the optoelectronic properties of halide perovskites. In particular, the extent of phonon damping—whether a mode is underdamped or overdamped—directly influences electron-phonon coupling and charge carrier behavior[34].

To further investigate this, the boundaries between underdamped and overdamped phonon behavior were analyzed across the composition range. This allows for the identification of regions where phonon damping transitions occur, providing insight into where dynamic fluctuations are most likely to affect optoelectronic properties.

A phonon mode is underdamped when its oscillatory behavior persists over multiple cycles before decaying, meaning the system retains a well-defined vibrational coherence. In contrast, a phonon mode is overdamped when its oscillations are suppressed, and the mode decays before completing a full cycle. This transition occurs when the damping rate exceeds the vibrational frequency, effectively transforming the phonon into a non-propagating relaxation mode.

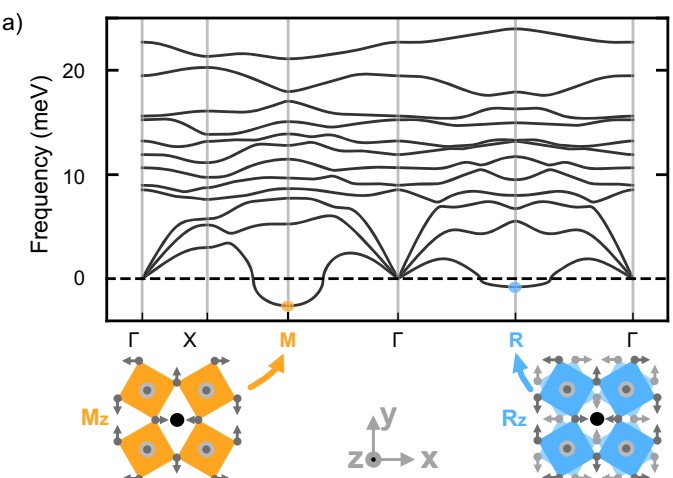
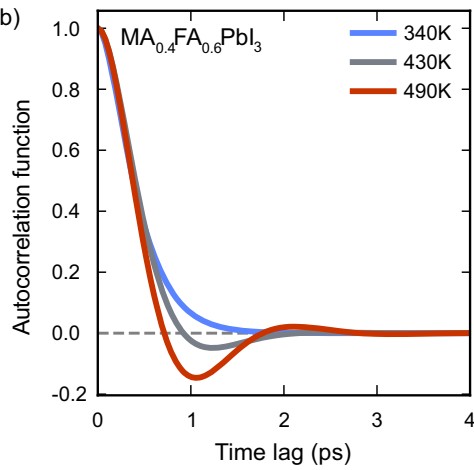

**Fig. 5 | Phonon dispersion and overdamping. a** Phonon dispersion of MAPbI$_3$ in the cubic phase. The displacement patterns corresponding to the imaginary modes at the M and R points (indicated by negative frequencies) are illustrated, with arrows representing the displacement of iodine atoms. In the M$_z$ mode, octahedral layers tilt uniformly in the same direction, whereas in the R$_z$ mode, adjacent layers tilt alternately in opposite directions (dark and light gray arrows). **b** The ACF of the R mode for MA$_{0.4}$FA$_{0.6}$PbI$_3$ at 340 K, 430 K and 490 K, illustrating the transition from under to overdamped dynamics with decreasing temperature.

In perovskites, phonon modes can become overdamped near structural phase transitions, where the potential energy surface flattens and vibrational modes soften. This has been observed in related materials such as CsPbBr$_3$, where overdamped phonons persist well above phase transition temperatures, leading to enhanced dynamic disorder and influencing charge transport properties[35,36]. Given that soft phonon modes enhance electron-phonon coupling, mapping overdamped and underdamped regions provides direct insight into where electronic properties—such as band structure renormalization, charge transport, and recombination rates—are most strongly affected by lattice dynamics.

To determine whether a phonon mode was underdamped or overdamped, its autocorrelation function (ACF) was calculated, and the decay characteristics were extracted. A damped harmonic oscillator (DHO) fit was applied to obtain the decay time and oscillation frequency, allowing classification of each mode. We note that our approach naturally captures the rotational dynamics of the organic cations, and these contributions are thus fully incorporated in the following analysis. A detailed analysis of cation dynamics in the FAPbI$_3$ system can be found in ref. 22 (also see Fig. S5).

The ACF fits for the R mode of MA$_{0.4}$FA$_{0.6}$PbI$_3$ can be seen for three different temperatures in Fig. 5. This illustrates how the oscillation vanishes when decreasing temperature, indicating a transition of the ACF from being an underdamped to overdamped oscillator. Extending this analysis across the entire composition range allowed for the identification of the boundaries between underdamped and overdamped regions in the phase diagram (Fig. 2d).

Three distinct dynamic regions emerged. At high temperatures, both M and R modes were underdamped, retaining vibrational coherence. At lower temperatures, both modes became fully overdamped, losing coherence due to strong phonon damping. An intermediate region was identified where only the M mode was overdamped, revealing an asymmetry in how phonon dynamics evolve with composition and temperature. Interestingly, there exists a specific composition at which the transition from underdamped to overdamped dynamics occurs at the same temperature. At this composition, the free energy landscapes of the M and R modes become nearly degenerate, allowing the system to dynamically fluctuate between in-phase and out-of-phase octahedral tilting. This composition of mode crossover coincides with the MPB, suggesting that the onset of phonon overdamping may serve as a hallmark of the MPB. Indeed, as shown above, the structure at the MPB exhibits a

mixture of layers with M and R tilting at very short periodicity (Fig. 4), indicating that the interface energy between these tilting patterns becomes vanishingly small.

The identification of overdamped and underdamped phonon regions has direct consequences for electronic transport and optical properties. One can expect regions where phonons become overdamped to often coincide with enhanced charge carrier interactions with lattice vibrations, leading to strong phonon anharmonicity and dynamic disorder. These effects can significantly impact band structure, carrier scattering, and defect tolerance. As halide perovskites are frequently used in solar cells and optoelectronic devices operating well above room temperature, understanding phonon dynamics at elevated temperatures is crucial for predicting device performance and stability. By mapping phonon damping transitions, this analysis provides a framework for anticipating temperature and composition-dependent changes in optoelectronic behavior, guiding future experiments and theoretical studies.

**Electronic structure**

To assess the influence of octahedral and cation dynamics on the electronic structure, we performed density functional theory (DFT) calculations on representative snapshots from NEP-MD simulations at 330 K, which inherently capture phonon dynamics and the rotational dynamics of the organic cations. This temperature was selected to represent room temperature while accounting for the offset between model and experiment for the cubic-tetragonal phase boundary, which amounts to ~30 K. To quantify the effective strength of electron-phonon coupling under these conditions—i.e., in metastable (dynamically stabilized) structures at finite temperature—we computed the standard deviation of the VBM and conduction band minimum (CBM) energies across the sampled structures as a function of FA content (Fig. 6).

In general, band edge fluctuations are more pronounced in mixed compositions than in pure phases, with the largest variations in the VBM and CBM occurring around 20% and 40% FA, respectively. These enhanced fluctuations suggest an increased potential for carrier localization, as variations in the electronic structure can create local energy minima that trap charges. Interestingly, this non-monotonic behavior of electron-phonon coupling emerges in the cubic phase near the mode crossover composition and, consequently, the MPB (Fig. 2d). Since these two phenomena are intrinsically linked, as discussed above, this suggests that systems exhibiting MPBs are also likely to

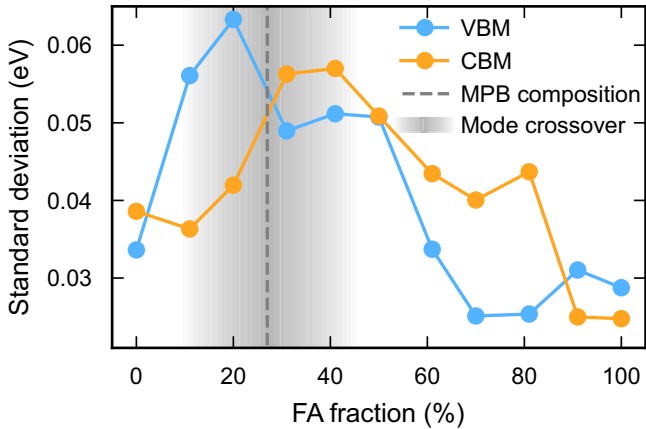

**Fig. 6 | Effective electron-phonon coupling strength.** Standard deviations of the valence band maximum (VBM) and conduction band minimum (CBM) levels as a function of FA composition at 330 K. The variation of the band edges becomes maximal in the range from 20 to 40% FA and thus in the vicinity of the morphotropic phase boundary (MPB).

show enhanced electron-phonon coupling–providing a structural mechanism for tuning optoelectronic properties.

## Discussion

In this study, we systematically mapped the phase diagram of $MA_{1-x}FA_xPbI_3$ using machine-learned interatomic potentials and MD simulations. Our results reveal the presence of a MPB at ~27% FA content, marking the transition between out-of-phase and in-phase octahedral tilt patterns. This MPB remains nearly invariant with temperature, highlighting its compositional stability and structural significance.

Phonon mode analysis demonstrated that M and R phonon modes exhibit distinct overdamping behavior, with a mode crossover composition aligning with the MPB. This transition is accompanied by the formation of nanoscale layered structures with alternating tilt patterns, indicative of vanishingly small interface energies between competing structural motifs. These findings provide a structural explanation for the phase coexistence observed in this region.

Furthermore, our results offer a systematic and consistent description of the phase behavior of $MA_{1-x}FA_xPbI_3$, helping to reconcile previous partial and sometimes conflicting experimental assessments of this system. By providing a detailed, atomistic-level understanding of structural transitions and phonon dynamics, our work complements experimental efforts that have faced challenges in fully resolving the phase relationships of mixed-cation perovskites.

We also explored the implications for electron-phonon interactions, finding that band edge fluctuations are more pronounced near the MPB, suggesting an enhancement of dynamic disorder and charge carrier localization. These results establish a direct link between phonon dynamics, phase behavior, and electronic structure, introducing MPBs as a structural tuning mechanism for optoelectronic properties.

Our findings also demonstrate that signatures of the MPB could be present in experimental measurements of electron-phonon driven properties. As the frequencies of the transition-driving phonon modes usually decrease near the transition, the resulting enhancement in electron-phonon coupling should become apparent, owing to the relation between coupling strength and phonon frequency. In the vicinity of a MPB, multiple modes are expected to exhibit this behavior simultaneously, potentially amplifying the effect.

By demonstrating that phonon overdamping serves as a hallmark of the MPB, our study provides insights into composition-driven strategies for tuning the stability and performance of perovskite

materials. This framework contributes to the rational design of high-performance perovskite solar cells, where fine-tuning structural dynamics can play a key role in optimizing optoelectronic properties. Given that the basic conditions for MPBs are also fulfilled in other systems, including, e.g., $MA_{1-x}FA_xPbBr_3$[37,38], one can anticipate similar behavior in these materials as well.

## Methods
### Machine-learned interatomic potential
We constructed a machine-learned interatomic potential based on the fourth-generation NEP framework[39,40] using the iterative procedure described in ref. 41 utilizing the GPUMD[39] and CALORINE packages[42]. The training set was initially composed of both systematically and randomly strained and scaled structures, based on ideal structures (see the Zenodo record for a database of all structures used). An initial model was trained on all available data.

The dataset was then augmented with structures from several iterations of active learning. To this end, we additionally trained an ensemble of five models by randomly splitting the available data into training and validation sets. The ensemble was used to estimate the model error. MD simulations were then carried out at a range of temperatures and pressures, considering compositions over the entire concentration range. The ensemble was used to select structures with a high uncertainty, quantified by the standard deviations of energies and forces over the ensemble. Subsequently, we computed energies, forces, and stresses via DFT for these configurations and included them when training the next-generation NEP model. In total, the training set comprised 986 structures, corresponding to a total of 179,018 atoms. Structures were generated and manipulated using the ASE[43] and HIPHIVE packages[44].

For the NEP model, radial and angular cutoffs were set to 8 Å and 4 Å, respectively, and angular descriptors included both three and four-body components. The neural network architecture consisted of 30 descriptor nodes in the input layer (5 radial, 25 angular) and one hidden layer with 30 fully connected neurons. The final model was trained using the separable neuroevolution strategy[45] for e6 generations, after which the loss, based on the root-mean-squared errors, was deemed converged (Fig. S1; also see Figs. S2 and S3 of the SI). The model performance is illustrated in Fig. 1 and Fig. S4.

### Density functional theory calculations
The NEP model was fitted using forces, energies, and virials obtained from DFT calculations, employing the SCAN+rVV10 exchange-correlation functional[46] as implemented in the Vienna ab-initio simulation package[47,48] using projector-augmented wave[49,50] setups with a plane wave energy cutoff of 520 eV. The Brillouin zone was sampled with automatically generated $\Gamma$-centered **k**-point grids with a maximum spacing of 0.15 Å$^{-1}$.

### Molecular dynamics simulations
Structures were constructed from a 96-atom cell, corresponding to a $2 \times 2 \times 2$ primitive cubic unit cell, comprising 24 iodine atoms, 8 lead atoms, and 12 MA or FA molecules. The initial unit cell symmetry, defined by iodine displacements, was set to cubic ($a^0a^0a^0$) for cooling simulations and orthorhombic ($a^-a^-c^+$) for heating simulations. To obtain the desired system size, the structure was replicated along its axes. Random molecular substitutions were performed to achieve the target $MAPbI_3$:$FAPbI_3$ ratio, ensuring no spatial correlation in the MA/FA distribution.

MD simulations were carried out using the GPUMD package with a time step of 0.5 fs. The simulations employed the NPT ensemble via a Langevin thermostat and a stochastic cell rescaling barostat[51]. To construct the phase diagram, heating and cooling simulations were conducted between 1 K and 400 K at rates ranging from 7 K ns$^{-1}$ to 15 K ns$^{-1}$.

To assess finite-size effects, we monitored the lattice parameters of pure MAPbI$_3$ and FAPbI$_3$ as a function of supercell size. These convergence tests showed that supercells with at least 12,000 atoms were sufficient to obtain reliable results. for phase diagram construction we therefore used supercells ranging from 12,000 to 96,000 atoms.

To examine the impact of octahedral and cation dynamics on the electronic structure, we performed MD simulations using the NEP model in a 768-atom supercell at 330 K with a fixed cell shape and volume. This temperature was chosen since the tetragonal-cubic phase boundary occurs at slightly higher temperatures than observed experimentally. We investigated MA/FA mixing ratios with FA fractions of 0%, 11%, 20%, 31%, 41%, 50%, 61%, 70%, 81%, and 100%. From each MD trajectory, we selected 20 snapshots and computed the electronic structure using DFT, sampling $k$-space only at the $\Gamma$ point while maintaining consistency with training runs. We then extracted the positions of the VBM and CBM to quantify their fluctuation magnitudes.

## Phonon modes

To classify the local crystal structure along a MD trajectory, we employed local phonon mode projection to quantify atomic displacements relative to the ideal cubic perovskite structure. The analysis focused on the M and R phonon modes, which correspond to the tilting of PbI$_6$ octahedra in the 96-atom unit cell. These modes are threefold degenerate, denoted as $M_{x/y/z}$ and $R_{x/y/z}$, depending on the tilt axis (see Fig. 5 for an illustration of $M_z$ and $R_z$). To quantify mode activation, the displacement of each iodine atom from the ideal cubic structure, $\mathbf{u}$, was projected onto the M and R mode eigenvectors, $\mathbf{e}_{M/R}$, yielding $Q_{M/R} = \mathbf{u} \cdot \mathbf{e}_{M/R}$. The resulting phonon mode activation was normalized by dividing $Q_{M/R}$ by the number of iodine atoms per unit cell. This normalization ensures that the activation represents the average iodine displacement along the phonon mode.

To determine the character of the phonon modes, we fitted their ACFs with a DHO model, characterized by a natural frequency $\omega_0$ and damping constant $\Gamma$. The fitting function is given by:

$$C_Q^{DHO}(t) = A^{-t/\tau}\left(\cos\omega_e t + \frac{\Gamma}{2\omega_e}\sin\omega_e t\right),$$

where the effective frequency is defined as $\omega_e = \sqrt{\omega_0^2 - \frac{\Gamma^2}{4}}$. The phonon mode is classified as underdamped if $\omega_e$ is real, and overdamped if $\omega_e$ is imaginary. That is in the underdamped case $\omega_0 > \Gamma/2$, while in the overdamped case $\omega_0 < \Gamma/2$.

## Data availability

The NEP model and the database of DFT calculations used to train this model have been deposited in the Zenodo database under accession code https://doi.org/10.5281/zenodo.14992798[52]. Source data are provided as a Source Data file. Source data are provided with this paper.

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

## Acknowledgements
We gratefully acknowledge funding from the Swedish Research Council (Nos. 2019-03993, 2020-04935, and 2021-05072), the Knut and Alice Wallenberg Foundation (Nos. 2023.0032 and 2024.0042), the Swedish Strategic Research Foundation (FFL21-0129), and the European Research Council (ERC Starting Grant No. 101162195) as well as computational resources provided by the National Academic Infrastructure for Supercomputing in Sweden at NSC, PDC, and C3SE partially funded by the Swedish Research Council through grant agreement No. 2022-06725, as well as the Berzelius resource provided by the Knut and Alice Wallenberg Foundation at NSC.

## Author contributions
T.H. carried out the MD simulations as well as their analysis with support from E.F. and S.D., and wrote the first draft of the manuscript. E.F. and P.E. constructed the NEP model. J.W. and P.E. carried out the DFT calculations and supervised the project. P.E. proposed the initial concept, with all authors contributing to its further development. All authors contributed to writing and reviewing the manuscript.

## Funding

## Competing interests
The authors declare no competing interests.
