## [Transparent Peer Review file · Nature Communications]

A Morphotropic Phase Boundary in MA(1-x)FA(x)PbI₃: Linking Structure, Dynamics, and Electronic Properties

Corresponding Author: Professor Paul Erhart

Version 0:

Reviewer comments:

Reviewer #1

(Remarks to the Author)

This study by Tobias Hainer et al. offers a comprehensive investigation of the MA_{1-x}FA_xPbI₃ mixed-cation perovskite system. By employing advanced molecular dynamics simulations based on a machine-learned interatomic potential, the authors successfully map out structural phase transitions across a range of FA concentrations. Notably, they identify a morphotropic phase boundary (MPB) at approximately 27% FA content, which separates regions characterized by distinct octahedral tilt patterns within the inorganic sublattice.

The work is timely, well-executed, and will likely be of interest to researchers working on hybrid perovskites and structural phase behavior. However, I believe several key aspects warrant further clarification or discussion:

i) The term morphotropic phase boundary is traditionally reserved for transitions between distinct crystallographic symmetries (e.g., tetragonal and rhombohedral phases). While the transition described here bears conceptual similarity, it is not clear that the term "MPB" is entirely appropriate for a boundary defined solely by changes in octahedral tilt patterns of tetragonal symmetries. The authors should address this point in the context of the conventional MPB definition used in inorganic perovskites and be more careful with the definitions, especially in the paper title.

ii) Are similar structural transitions or tilt-induced boundaries expected in related hybrid perovskite systems, such as MA_{1-x}FA_xPbBr₃, which have been extensively studied experimentally (e.g., 10.1021/acsami.7b06001; 10.1021/acs.chemmater.1c00885)? A comparison could strengthen the generality and relevance of the findings.

iii) The layered structure observed at the MPB (Figure 4) is particularly intriguing. Could the authors elaborate on potential finite-size effects or boundary conditions in their simulations that might influence the formation or stability of these layers?

iv) In conventional inorganic perovskites, the dielectric susceptibility often peaks near the MPB due to enhanced polarizability. However, previous studies (e.g., Mohanty et al.) do not report such behavior in similar hybrid systems. The authors are encouraged to comment on whether an enhanced dielectric response is expected in their case and, if not, to provide possible explanations.

v) The behavior of the molecular cations (MA and FA) at the MPB remains unexplored in this study. Given that the authors' simulation framework incorporates the organic sublattice, it would be valuable for them to discuss how MA and FA dynamics may contribute to or be affected by the phase boundary.

vi) The choice of 330 K for the electronic structure analysis (Figure 6) should be justified. How does this temperature relate to the identified MPB and the phonon overdamping regions shown in the phase diagram?

vii) While the computational analysis is rigorous, the manuscript would benefit from a more detailed discussion directed at experimentalists, particularly given the broad and interdisciplinary readership of Nature Communications. The authors should consider offering practical suggestions or identifying experimental signatures that could help detect the proposed MPB, especially in light of discrepancies reported in the literature.

Reviewer #2

(Remarks to the Author)

Referee report on Ms. NCOMMS-25-19253 entitled "A Morphotropic Phase Boundary in $MA_{1-x}FA_xPbI_3$: Linking Structure, Dynamics, and Electronic Properties" by T. Hainer et al. for consideration for Nature Communications.

This is without doubt a very nice piece of work that I enjoyed much reading. The manuscript is very well and elegantly written, providing much physical insight about the relationship between phonon dynamics, structural phase behavior and electronic properties in a nowadays very relevant semiconductor system, namely mixed-cation $MA_{1-x}FA_xPbI_3$ perovskites. Here the composition-temperature phase diagram is mapped out using machine-learning methods to obtain interatomic potentials for the molecular dynamics simulations. The main finding of this work is a morphotropic phase boundary (MPB) between phases characterized by out-of-phase and in-phase octahedral tilting patterns at a composition close to 30% FA content. This leads to the formation of stripe domains (ca. 20 nm thick) alternating both mentioned phases, which might explain some contradictory or inconclusive experimental results of the literature. In addition, density functional theory calculations were used to show that, near the MPB, band-edge fluctuations exhibit a maximum, indicating an enhancement of electron-phonon interactions and dynamic disorder effects.

To be honest, I don't have any technical criticism, for the simulations and calculations were performed with great care and employing state of the art methodologies. However, there are a few points which might need further clarification, mainly regarding the phonon calculations.

1. The first issue that needs consideration is the impact of the negative phonon frequencies at the M and R points, which is taken for the evidence of soft-mode behavior. Of course, this is true close to a phase transition triggered by structural instabilities related to certain phonon modes which become soft (frequency goes to zero). The point is that this happens already for $MAPbI_3$ in the cubic phase and for temperatures where the structure is perfectly stable. To my understanding this is typically due to shortcomings in the calculations due to the (small) finite size of the super-cell been considered, which leads to non-converged results (I fully understand that the super-cell cannot be enlarged as much as desired for the unaffordable calculation costs). Normally, this is not so important if happens in a small range outside the point, as in the present case. However, since the conclusions depend so much on such soft-mode behavior, I wonder if the authors can convincingly argue that this does not pose a problem.
2. The other point which is for me really intriguing and might be related to the previous point, is why M and R phonon modes are overdamped at "low" temperatures but underdamped at "high" temperatures. The damping of a phonon mode is just given by its homogeneous broadening, which is inverse proportional to its lifetime. The latter is mainly determined by anharmonicities stemming from (higher-order) phonon-phonon interactions. These are simply more important the higher the temperature (much larger phonon occupation numbers). Thus, what am I missing here? In fact, all Γ -point phonon linewidths become substantially smaller at low temperatures, as measured by Raman (see, for instance, Ref. [18] and the work of Leguy et al., *Phys. Chem. Chem. Phys.* **2016**, 18, 27051-27066). I agree with the authors that an elegant way to check if a phonon mode is over or underdamped is to look at the autocorrelation function. However, I cannot refrain to think of this as an artifact of the nonconverged calculations. Negative frequencies means that the eigenvalues are still complex numbers. The non-vanishing imaginary part certainly impacts the mode linewidth (lifetime), i.e. the magnitude of the damping...
3. In my opinion, the implications of the enhancement of electron-phonon interactions around MPB concentrations are a bit of a hype. Is there any evidence of the effects of enhanced electron-phonon couplings on the electronic structures of the mixed-cation perovskites? It doesn't seem like that. For example, an inspection of the data from Ref. [18] does not show any unusual behavior of the band gap or its temperature dependence in the MPB concentration range...

In conclusion, I strongly recommend publication of this article in Nat. Commun. after my comments and suggestions have been considered.

Reviewer #3

(Remarks to the Author)

This manuscript (ID# NCOMMS-25-19253) entitled "A Morphotropic Phase Boundary in $MA_{1-x}FA_xPbI_3$: Linking Structure, Dynamics, and Electronic Properties" by Hainer et al. probes mixed halide perovskites via detailed theoretical investigations of the evolution of the structure as a function of the composition. The theoretical molecular dynamics simulations are based on machine-learned interatomic potentials, and the authors present a structure-composition phase diagram for this family of compounds, while identifying a morphotropic phase boundary (MPB) at approximately 27% FA content. The authors refer to the experimentally obtained phase diagram of this series, underlining some of the conflicting claims, thereby motivating the need for the present theoretical investigation. The authors analyze the phonons to identify structural subtleties. They also report enhanced band-edge fluctuations as the system approaches the morphotropic phase boundary, suggesting an increased electron-phonon coupling and possibilities of influencing various properties of these materials.

I have several reservations about this work that make me doubt the validity of the conclusions. While it is indeed true that there are some specific disagreements in the experimentally reported phase diagram of this family of compounds, the origin of such disagreements most likely arises from the different techniques, at times very indirect ones, used to conclude

structural information, instead of using direct structural probes. The authors of this manuscript are also sensitive to this aspect, as evident in their comment on some of the results they have cited in this manuscript, stating, "Note, however, that these assignments are not based on crystallographic analysis." The real question here is whether structure-composition phase diagrams can be reliably extracted from theoretical calculations, particularly for this class of compounds, so that these may be used to correct structural information experimentally obtained using direct structural probes, such as diffraction techniques, as attempted here. Theoretical calculations have important roles to play in helping us understand microscopic interactions and the origins of different phenomena; however, revising experimental results will require substantiating theoretical insights with solid experimental proof. This is particularly so for the class of compounds investigated here, but, unfortunately, experimental substantiation of the theoretical claims is entirely missing in this work.

The entire class of compounds is well-known to have phonon instabilities at the R and M points, as also reported by these authors in Fig. 5a. These instabilities, while being clear evidence of the inadequacies of the electronic structure calculations for these compounds at all compositions, arise from rotational degrees of freedom of the MA and FA ions that are not part of the theoretical framework used here. The rotational degrees of freedom of these ions have profound impacts on many physical properties of these compounds; specifically, an order of magnitude larger dipole moment on MA affects the lattice dynamics in a way that is considerably different from that of the FA ions. The limitation of the theoretical approach is also evident in the fact that the tetragonal-orthorhombic phase transition in MAPbI₃ is not captured on the cooling cycle, while experimentally, the first-order phase transition across the T-O boundary shows up prominently in both heating and cooling runs of many experimental probes, such as differential scanning calorimetry, and dielectric properties. For FAPbI₃, the phase transition between one of the two tetragonal structures and the orthorhombic structure did not show up on the cooling run, while the other tetragonal structure to the orthorhombic structure exhibited the phase transition on both heating and cooling runs. This is strange, since energetically both tetragonal structures should have the same propensity to transform into the orthorhombic structure. DFT calculations presented later in the text are of limited validity since such calculations cannot account for the rotational degrees of freedom of the MA and FA ions, and the systems show phonon instabilities, making most conclusions somewhat speculative.

Version 1:

Reviewer comments:

Reviewer #1

(Remarks to the Author)

The response from the Authors is solid. I recommend for publication.

Reviewer #2

(Remarks to the Author)

In my opinion, the authors have adequately answered all the questions posed by the three reviewers. I would like to add that I do not share the third reviewer's reservations about this work, and even less do I doubt the validity of the conclusions. Indeed, I find the good agreement between the experimentally determined and calculated electron-phonon coupling strengths in mixed-cation perovskites extremely suggestive (new Fig. S6 in the Supplementary Information), for instance.

In conclusion, this is an excellent and solid work, well written for a broad audience, and one that will have a major impact on the scientific community; therefore, I strongly recommend its publication in Nature Communications in its current version.

Reply To Referee 1

Referee:

This study by Tobias Hainer et al. offers a comprehensive investigation of the $MA_{1-x}FA_xPbI_3$ mixed-cation perovskite system. By employing advanced molecular dynamics simulations based on a machine-learned interatomic potential, the authors successfully map out structural phase transitions across a range of FA concentrations. Notably, they identify a morphotropic phase boundary (MPB) at approximately 27% FA content, which separates regions characterized by distinct octahedral tilt patterns within the inorganic sublattice. The work is timely, well-executed, and will likely be of interest to researchers working on hybrid perovskites and structural phase behavior. However, I believe several key aspects warrant further clarification or discussion:

We thank the Referee for the positive comments.

Referee:

i) The term morphotropic phase boundary is traditionally reserved for transitions between distinct crystallographic symmetries (e.g., tetragonal and rhombohedral phases). While the transition described here bears conceptual similarity, it is not clear that the term "MPB" is entirely appropriate for a boundary defined solely by changes in octahedral tilt patterns of tetragonal symmetries. The authors should address this point in the context of the conventional MPB definition used in inorganic perovskites and be more careful with the definitions, especially in the paper title.

We appreciate the referee's point and agree that greater care is warranted in our use of the term morphotropic phase boundary (MPB).

A MPB describes a region in a solid solution where the crystal structure changes abruptly with small changes in composition. It thus refers to a structural transformation caused by changing the chemical makeup of a material rather than its temperature. While it is usually used in the context of ferroelectric or piezoelectric materials, the basic phenomenon is not limited to such materials. In order for a MPB to be possible, one thus requires two end members with different symmetries that are fully miscible. These requirements are fulfilled not only in $PbTi_{1-x}Zr_xO_3$ (PZT) but also $MA_{1-x}FA_xPbI_3$. Specifically, $MAPbI_3$ and $FAPbI_3$ have different symmetries in the temperature range of interest, and our model predicts full miscibility (which is consistent with experimental observations). The two phases involved here are both tetragonal but nonetheless have different space groups.

To provide a more pedagogical introduction and better context, we have integrated this information in the sixth paragraph of the introduction, which now reads:

Modification:

A key observation is that the tetragonal phases of $MAPbI_3$ and $FAPbI_3$, which occur at room temperature, exhibit opposing tilt patterns ($a^0a^0c^-$ vs $a^0a^0c^+$) and belong to two different space groups ($I4/mcm$ vs $P4/mbm$). Given the full miscibility of $FAPbI_3$ and $MAPbI_3$, this suggests the presence of at least one MPB. In general, a MPB is a region in the composition-temperature phase diagram of a solid solution — typically in but not limited to ferroelectric or piezoelectric materials — where two crystallographic phases with different symmetries coexist or are nearly degenerate in energy.

Referee:

ii) Are similar structural transitions or tilt-induced boundaries expected in related hybrid perovskite systems, such as $MA_{1-x}FA_xPbBr_3$, which have been extensively studied experimentally (e.g., 10.1021/ac-sami.7b06001; 10.1021/acs.chemmater.1c00885)? A comparison could strengthen the generality and relevance of the findings.

Since the criteria for the presence of a MPB are general, they can also be expected in other perovskite systems. So, if the boundary phases have different space groups and the system is completely miscible one can expect there to be at least one MPB in the phase diagram. These conditions are also fulfilled, e.g., in the $MA_{1-x}FA_xPbBr_3$ system. We added the following sentence in the conclusions section:

Modification:

Given that the basic conditions for MPBs are also fulfilled in other systems, including, e.g., $MA_{1-x}FA_xPbBr_3$ [Ono (2017), Šimėnas (2021)], one can anticipate similar behavior in these materials as well.

Referee:

iii) The layered structure observed at the MPB (Figure 4) is particularly intriguing. Could the authors elaborate on potential finite-size effects or boundary conditions in their simulations that might influence the formation or stability of these layers?

There are two key points to address in response to this comment.

First, we have previously observed a noticeable difference between the smaller and larger cells in terms of the final domain pattern. This is evident when comparing Figs. 4a and b: Figure 4a displays a one-dimensional layered structure, which appears when using smaller simulation cells, while Fig. 4b reveals the emergence of diagonal stripe patterns that appear only in larger cells. We observe that the latter behavior is due to the simultaneous nucleation of several domains with different orientations that subsequently grow into each other, leading to the formation of boundaries. When this occurs, we obtain 90° domain boundaries in our structure. This type of event is more likely to occur the larger the cell. Classical nucleation theory implies that the frequency of these nucleation events is also sensitive to the degree of undercooling, i.e., the difference between the current temperature and the equilibrium transition temperature. One can expect that for very long time scales the density of these boundaries decreases as the system can lower its (free) energy by removing these defects. However, these time scales appear to be much longer than those accessible in our simulations.

Second, one can raise the question regarding whether these stripes of different tilt direction have a size dependence. The above logic suggests that on the time scale accessible by our simulations, the domain size is primarily determined by the nucleation rate, which (for sufficiently large cells) should depend on the degree of undercooling (as argued above). To establish that the cells used previously are “sufficiently large” in this sense, we carried out new simulations for cells with as many as 2.6 million atoms. These simulations resulted in patterns with very similar widths. Notably, we obtained a distribution of domain widths, and the different orientations were not equally distributed in the thus obtained structures. Both of these observations are consistent with the nucleation and growth picture described above.

To incorporate these aspects into the paper we added the following text in the fifth paragraph of the section on the “Character of the morphotropic phase boundary”:

Modification:

The formation of different domains should not be interpreted as a finite-size effect, but rather as a consequence of the stochastic nature of nucleation pathways at finite cooling rates, and thus a finite-time effect. The coexistence of regions with different orientations primarily reflects the competition between local energy barriers for reorientation and the finite time available for domain wall motion, rather than a direct effect of system size itself. This is supported by the fact that for sufficiently large cells, one cannot observe a correlation between the size of the domains and the size of the simulation cell.

Referee:

iv) In conventional inorganic perovskites, the dielectric susceptibility often peaks near the MPB due to enhanced polarizability. However, previous studies (e.g., Mohanty et al.) do not report such behavior in similar hybrid systems. The authors are encouraged to comment on whether an enhanced dielectric response is expected in their case and, if not, to provide possible explanations.

This is a very good question that we asked ourselves as well. In systems such as $\text{PbTi}_{1-x}\text{Zr}_x\text{O}_3$ (PZT) the two phases that meet at the MPB that is located at $x \approx 0.45$ are both ferroelectric and thus they feature a spontaneous polarization (and thus permanent dipoles). Our system, however, shows anti-ferrodistortive behavior, which involves non-polar structural distortions (alternating octahedral tilts) that do lead to a zero net polarization (no permanent dipoles). We therefore do not expect a strong signature of the MPB in the dielectric susceptibility.

To incorporate these aspects into the paper, we added the following text at the end of the section on the “Character of the morphotropic phase boundary”:

Modification:

One should, however, note that in contrast to systems with well-established MPBs such as $\text{PbTi}_{1-x}\text{Zr}_x\text{O}_3$, in the present case the two phases that meet at the MPB exhibit anti-ferrodistortive behavior and thus no spontaneous polarization. As a result, we do not expect a strong signature of the MPB in the dielectric response.

Referee:

v) The behavior of the molecular cations (MA and FA) at the MPB remains unexplored in this study. Given

that the authors' simulation framework incorporates the organic sublattice, it would be valuable for them to discuss how MA and FA dynamics may contribute to or be affected by the phase boundary.

We agree with the referee that the MA and FA effects around the phase boundary would be a relevant addition to the paper. However, throughout our study we did not observe any significant changes in the cation behavior, whether in terms of orientational ordering, reorientation times, or spatial correlations across the phase transition. For this reason, and to maintain focus on the key findings, we chose not to include a detailed discussion of the cations in the manuscript. To highlight that this aspect has been investigated, we have included Fig. S5 in the Supporting Information.

Figure S5: Direction of MA and FA bonds as a function of composition. Bond distribution for C-N and C-H in MA, as well as N-N and C-H in FA at 10% and 90% FA fraction.

Referee:

vi) The choice of 330 K for the electronic structure analysis (Figure 6) should be justified. How does this temperature relate to the identified MPB and the phonon overdamping regions shown in the phase diagram?

Our goal is to demonstrate that the overdamping of both M and R-modes and in particular the crossing near 27% FA (which is connected to the MPB at lower temperatures) has a noticeable effect on the behavior near room temperature and above the cubic-tetragonal phase boundary. Since the model systematically overestimates this transition by about 30 K, we therefore used a temperature of 330 K.

We have added the following text to the manuscript:

Modification:

To assess the influence of octahedral and cation dynamics on the electronic structure, we performed DFT calculations

on representative snapshots from NEP-MD simulations at 330 K, which inherently capture phonon dynamics and the rotational dynamics of the organic cations. This temperature was selected to represent room temperature while accounting for the offset between model and experiment for the cubic–tetragonal phase boundary, which amounts to approximately 30 K.

Referee:

vii) While the computational analysis is rigorous, the manuscript would benefit from a more detailed discussion directed at experimentalists, particularly given the broad and interdisciplinary readership of Nature Communications. The authors should consider offering practical suggestions or identifying experimental signatures that could help detect the proposed MPB, especially in light of discrepancies reported in the literature.

We agree with the Referee that additional context for experimentalists would be valuable. To address this, we have extended the discussion to clarify under which conditions we expect MPBs to appear, along with brief guidance on how it may be identified experimentally. The following section has been included as the final paragraph of the “Character of the morphotropic phase boundary” (note that parts of this text are repeated from above):

Modification:

In general, when aiming to experimentally investigate MPBs, whether in $MA_{1-x}FA_xPbI_3$ or in other systems, it is important to recall the conditions necessary for their formation. Specifically, there must be distinct crystallographic phases at the boundary compositions as well as complete miscibility. With these criteria in mind, one can assess whether a given material is likely to exhibit a MPB. If so, this can serve as a basis for exploring possible domain structures and the coexistence of phases, as our work suggests that this could be good indicators for MPBs. Beyond structural considerations, we can also discuss strong electron-phonon coupling. The presence of soft phonon modes generally enhances the electron-phonon interaction, as the coupling strength scales as $1/\sqrt{\omega}$, where ω is the phonon frequency. This implies that the coupling should be stronger near a MPB, where multiple modes exhibit such soft behavior. As a result, the influence of the MPB on the electron-phonon interaction is expected to be significant. This is analyzed below in the section on electronic structure. Experimentally probing this behavior could thus provide further insights into the impact of MPBs on materials performance. One should, however, note that in contrast to systems with well-established MPBs such as $PbTi_{1-x}Zr_xO_3$, in the present case the two phases that meet at the MPB exhibit anti-ferrodistortive behavior and thus no spontaneous polarization. As a result, we do not expect a strong signature of the MPB in the dielectric response.

Reply To Referee 2

Referee:

This is without doubt a very nice piece of work that I enjoyed much reading. The manuscript is very well and elegantly written, providing much physical insight about the relationship between phonon dynamics, structural phase behavior and electronic properties in a nowadays very relevant semiconductor system, namely mixed-cation $MA_{1-x}FA_xPbI_3$ perovskites. Here the composition-temperature phase diagram is mapped out using machine-learning methods to obtain interatomic potentials for the molecular dynamics simulations. The main finding of this work is a morphotropic phase boundary (MPB) between phases characterized by out-of-phase and in-phase octahedral tilting patterns at a composition close to 30% FA content. This leads to the formation of stripe domains (ca. 20 nm thick) alternating both mentioned phases, which might explain some contradictory or inconclusive experimental results of the literature. In addition, density functional theory calculations were used to show that, near the MPB, band-edge fluctuations exhibit a maximum, indicating an enhancement of electron-phonon interactions and dynamic disorder effects.

We thank the Referee for the positive comments.

Referee:

To be honest, I don't have any technical criticism, for the simulations and calculations were performed with great care and employing state of the art methodologies. However, there are a few points which might need further clarification, mainly regarding the phonon calculations.

Referee:

1. The first issue that needs consideration is the impact of the negative phonon frequencies at the M and R

points, which is taken for the evidence of soft-mode behavior. Of course, this is true close to a phase transition triggered by structural instabilities related to certain phonon modes which become soft (frequency goes to zero). The point is that this happens already for MAPbI₃ in the cubic phase and for temperatures where the structure is perfectly stable. To my understanding this is typically due to shortcomings in the calculations due to the (small) finite size of the super-cell been considered, which leads to non-converged results (I fully understand that the super-cell cannot be enlarged as much as desired for the unaffordable calculation costs). Normally, this is not so important if happens in a small range outside the Γ point, as in the present case. However, since the conclusions depend so much on such soft-mode behavior, I wonder if the authors can convincingly argue that this does not pose a problem.

It is indeed well established that materials such as MAPbI₃ and FAPbI₃ exhibit imaginary phonon modes in their ideal cubic structures at zero Kelvin. These imaginary modes, particularly at the M and R points, indicate structural instabilities that, if followed, would drive the material toward lower-symmetry, distorted phases at low temperatures (see, e.g., <http://doi.org/10.1038/s42005-023-01297-8>). However, slightly above room temperature, these perovskites adopt a metastable cubic phase, i.e., that the phase is dynamically stabilized. This means that during molecular dynamics (MD) simulations in the cubic phase, the system fluctuates along the directions of the M and R modes, which shows up as phonon activation. Vibrational entropy thus hinders the system from falling into these lower symmetry states.

In regards to the comment about size effects. Size effects are mainly a concern when performing these calculations using density functional theory (DFT) on smaller supercells. Here, we employ large-scale MD simulations that are well converged in regards to the system size. It can be noted that we have carefully investigated finite-size (and time) effects in inorganic halide perovskites in earlier work, see <http://doi.org/10.1021/acs.jpcc.3c01542>.

Referee:

2. The other point which is for me really intriguing and might be related to the previous point, is why M and R phonon modes are overdamped at "low" temperatures but underdamped at "high" temperatures. The damping of a phonon mode is just given by its homogeneous broadening, which is inverse proportional to its lifetime. The latter is mainly determined by anharmonicities stemming from (higher-order) phonon-phonon interactions. These are simply more important the higher the temperature (much larger phonon occupation numbers). Thus, what am I missing here? In fact, all Γ -point phonon linewidths become substantially smaller at low temperatures, as measured by Raman (see, for instance, Ref. [18] and the work of Leguy et al., Phys. Chem. Chem. Phys. 2016, 18, 27051-27066). I agree with the authors that an elegant way to check if a phonon mode is over or underdamped is to look at the autocorrelation function. However, I cannot refrain to think of this as an artifact of the nonconverged calculations. Negative frequencies means that the eigenvalues are still complex numbers. The non-vanishing imaginary part certainly impacts the mode linewidth (lifetime), i.e. the magnitude of the damping...

Indeed, it is generally expected that phonon-phonon interactions become stronger at higher temperatures due to increased phonon occupation, leading to enhanced anharmonicity and thus shorter phonon lifetimes (larger damping). However, for the M and R point modes analyzed here, the situation is more nuanced. These modes become dynamically unstable as the system crosses their respective phase transition temperatures, which drives the structural transformation, for example from the cubic to the tetragonal phase upon cooling. As a result, their dynamics can deviate from the standard anharmonic broadening picture close to the transition temperature T_c . This has been explicitly shown in previous studies of M and R modes in inorganic halide perovskites, see in particular Fig. 4 in Fransson et al. (2023) <https://doi.org/10.1038/s42005-023-01297-8> where the lifetimes increase slightly with temperature (lower damping at higher temperature).

Regarding the comment on convergence, the simulation cells are sufficiently large and the simulation times are long enough to ensure convergence. For instance, the fluctuations occur on the scale of a few picoseconds, whereas the simulations span nanoseconds.

To clarify how we worked with the autocorrelation function (ACF) we added the following text to the subsection "Phonon modes" in the "Methods" section:

Modification:

To determine the character of the phonon modes, we fitted their ACF with a damped harmonic oscillator (DHO)

model, characterized by a natural frequency ω_0 and damping constant Γ . The fitting function is given by:

$$C_Q^{DHO}(t) = A^{-t/\tau} \left(\cos \omega_e t + \frac{\Gamma}{2\omega_e} \sin \omega_e t \right),$$

where the effective frequency is defined as $\omega_e = \sqrt{\omega_0^2 - \frac{\Gamma^2}{4}}$. The phonon mode is classified as underdamped if ω_e is real, and overdamped if ω_e is imaginary. This corresponds to the conditions:

1. Underdamped: $\omega_0 > \Gamma/2$
2. Overdamped: $\omega_0 < \Gamma/2$

Referee:

3. In my opinion, the implications of the enhancement of electron-phonon interactions around MPB concentrations are a bit of a hype. Is there any evidence of the effects of enhanced electron-phonon couplings on the electronic structures of the mixed-cation perovskites? It doesn't seem like that. For example, an inspection of the data from Ref. [18] does not show any unusual behavior of the band gap or its temperature dependence in the MPB concentration range...

We agree with the reviewer that the referenced article does not report any unusual behavior in the band gap around the MPB, based on their photoluminescence measurements. However, our claim is not based on a direct change in the band gap, but rather on enhanced fluctuations of the electronic bands.

In this context, we would like to highlight Fig. S4 in the Supporting Information of the paper by Francisco-Lopez et al. (2020), which shows the electron-phonon coupling as a function of composition at 160 K. Notably, there is an increased amplitude near the MPB composition, suggesting stronger electron-phonon interactions in this regime. While a direct comparison is not possible due to the different temperatures, the general composition dependence is strikingly similar.

Both simulations and experiments yield a rather broad feature around the MPB. Our analysis allows us to connect this behavior to the MPB, see the figure below (which is also included in the SI as Fig. S5) for this comparison.

Effective electron-phonon coupling strength compared to experiment. The experimental data are taken Ref. [Francisco-López (2020)]. There is a clear correlation between the fluctuations of the valence band maximum (VBM) and conduction band minimum (VBM) obtained from DFT and the experimentally extracted electron-phonon coupling.

Referee:

In conclusion, I strongly recommend publication of this article in Nat. Commun. after my comments and suggestions have been considered.

We thank the Referee for their very positive overall assessment.

Reply To Referee 3

Referee:

This manuscript (ID# NCOMMS-25-19253) entitled “A Morphotropic Phase Boundary in MA1-xFAxPbI3:

Linking Structure, Dynamics, and Electronic Properties” by Hainer et al. probes mixed halide perovskites via detailed theoretical investigations of the evolution of the structure as a function of the composition. The theoretical molecular dynamics simulations are based on machine-learned interatomic potentials, and the authors present a structure-composition phase diagram for this family of compounds, while identifying a morphotropic phase boundary (MPB) at approximately 27% FA content. The authors refer to the experimentally obtained phase diagram of this series, underlining some of the conflicting claims, thereby motivating the need for the present theoretical investigation. The authors analyze the phonons to identify structural subtleties. They also report enhanced band-edge fluctuations as the system approaches the morphotropic phase boundary, suggesting an increased electron-phonon coupling and possibilities of influencing various properties of these materials.

Referee:

I have several reservations about this work that make me doubt the validity of the conclusions. While it is indeed true that there are some specific disagreements in the experimentally reported phase diagram of this family of compounds, the origin of such disagreements most likely arises from the different techniques, at times very indirect ones, used to conclude structural information, instead of using direct structural probes. The authors of this manuscript are also sensitive to this aspect, as evident in their comment on some of the results they have cited in this manuscript, stating, “Note, however, that these assignments are not based on crystallographic analysis.” The real question here is whether structure-composition phase diagrams can be reliably extracted from theoretical calculations, particularly for this class of compounds, so that these may be used to correct structural information experimentally obtained using direct structural probes, such as diffraction techniques, as attempted here. Theoretical calculations have important roles to play in helping us understand microscopic interactions and the origins of different phenomena; however, revising experimental results will require substantiating theoretical insights with solid experimental proof. This is particularly so for the class of compounds investigated here, but, unfortunately, experimental substantiation of the theoretical claims is entirely missing in this work.

We agree that experimental validation is the cornerstone for establishing phase diagrams. Yet in the case of $\text{MA}_{1-x}\text{FA}_x\text{PbI}_3$, several experimental studies using different techniques have not (yet) been able to generate a cohesive picture of the phase diagram, indicating the need for complementary approaches. In our work, our aim is therefore to complement existing experimental interpretations by offering theoretical and computational insights that help explain the observed results.

We acknowledge the referee’s concern that theoretical results alone cannot definitively resolve ambiguities in experimental phase diagrams. However, we would like to emphasize that our theoretical predictions are in agreement with a number of independent experimental observations and tie together results from different sources to a coherent phase diagram. In this context, we would also like to highlight the consistency between the calculated standard deviations of the VBM and VBM and the electron-phonon coupling reported by Francisco-López et al. (2020), as shown in the newly added Fig. S5 (also see our response to point 3 of Referee 2). Such agreement is non-trivial and provides further support for our conclusions.

Referee:

The entire class of compounds is well-known to have phonon instabilities at the R and M points, as also reported by these authors in Fig. 5a. These instabilities, while being clear evidence of the inadequacies of the electronic structure calculations for these compounds at all compositions, arise from rotational degrees of freedom of the MA and FA ions that are not part of the theoretical framework used here. The rotational degrees of freedom of these ions have profound impacts on many physical properties of these compounds; specifically, an order of magnitude larger dipole moment on MA affects the lattice dynamics in a way that is considerably different from that of the FA ions. The limitation of the theoretical approach is also evident in the fact that the tetragonal-orthorhombic phase transition in MAPbI_3 is not captured on the cooling cycle, while experimentally, the first-order phase transition across the T-O boundary shows up prominently in both heating and cooling runs of many experimental probes, such as differential scanning calorimetry, and dielectric properties. For FAPbI_3 , the phase transition between one of the two tetragonal structures and the orthorhombic structure did not show up on the cooling run, while the other tetragonal structure to the orthorhombic structure exhibited the phase transition on both heating and cooling runs. This is strange, since energetically both tetragonal structures should have the same propensity to transform into the orthorhombic structure. DFT calculations presented later in the text are of limited validity since such calculations cannot account for the rotational degrees of freedom of the MA and FA ions, and the systems

show phonon instabilities, making most conclusions somewhat speculative.

The reviewer’s comment regarding the treatment of rotational degrees of freedom for the MA and FA ions calls for clarification.

Phonon instabilities should not be considered the result of the inadequacy of electronic structure calculations per se but merely indicate the instability of a structure in the zero-temperature (harmonic) limit. Phonon instabilities are common in organic as well as inorganic perovskites and not exclusive to materials with organic cations.

At finite temperatures, anharmonicity (i.e., phonon-phonon interactions) can lead to stabilization of structures that are unstable in the zero-temperature limit as shown for inorganic perovskites, e.g., in <http://doi.org/10.1038/s42005-023-01297-8>. Under these conditions, the normal modes of the harmonic phonon dispersion are still very useful as they provide a complete and orthogonal basis for mapping and understanding the dynamics of the systems, which is the approach taken in Figure 5.

In the present approach, the machine-learned interatomic potential (MLIP) provides an accurate mapping of the DFT energy landscape to a computationally much more efficient model that enables us to comprehensively sample both the composition and temperature axes of the phase diagram. In doing so, we naturally sample the full configuration space of the perovskite structure, including the rotational dynamics of the organic cations. As such, the rotational degrees of freedom of MA and FA are naturally accounted for in our approach. This is now highlighted in the manuscript. We added the following text in the section “Phonon dynamics”:

Modification:

We note that our approach naturally captures the rotational dynamics of the organic cations, and these contributions are thus fully incorporated in the following analysis. A detailed analysis of cation dynamics in the FAPbI₃ system can be found in Ref. [Dutta (2005)] (also see Fig. S5).

The comment concerning the two transitions between tetragonal and orthorhombic symmetries prompts us to clarify the distinction between the two tetragonal phases. These phases differ in their octahedral tilt patterns, described in Glazer notation as $a^0a^0c^-$ and $a^0a^0c^+$, while the low-temperature orthorhombic structure adopts the $a^-a^-c^+$ tilt pattern. As a result, the transition from the $a^0a^0c^-$ tetragonal phase is associated with a sizable free energy barrier, while the transition from $a^0a^0c^+$ is not. Note that there are orthorhombic phases that could be accessed from the $a^0a^0c^-$ tetragonal structure through MD simulations; those are, however, not relevant for this system. A detailed analysis of these tilt patterns and the associated free energy landscape can be found in <http://doi.org/10.1021/acs.chemmater.3c01740>.

We added a brief comment about the energy barrier associated with switching from in-phase to out-of-phase tilting in the second paragraph of the section “Phase diagram from mode projection”:

Modification:

This barrier is attributed to the required change in the tilt pattern of the third axis, from in-phase to out-of-phase [Fransson (2023b)].

We also add a clarifying statement regarding why FAPbI₃ shows similar behavior on heating and cooling in the third paragraph of the section “Phase diagram from mode projection”:

Modification:

In contrast, FAPbI₃ shows nearly identical behavior during heating and cooling (Fig. 2e–g), as the sampled tetragonal-to-orthorhombic transition occurs between phases that are energetically more similar.